# More Adult Women than Men at High Cardiometabolic Risk Reported Worse Lifestyles and Self-Reported Health Status in the COVID-19 Lockdown

**DOI:** 10.3390/nu16132000

**Published:** 2024-06-24

**Authors:** Alejandro Oncina-Cánovas, Laura Compañ-Gabucio, Jesús Vioque, Miguel Ruiz-Canela, Dolores Corella, Jordi Salas-Salvadó, Montserrat Fitó, Alfredo Martínez, Ángel M. Alonso-Gómez, Julia Wärnberg, Dora Romaguera, José López-Miranda, Ramón Estruch, Francisco J. Tinahones, José Lapetra, Jacqueline Álvarez-Pérez, Aurora Bueno-Cavanillas, Josep A. Tur, Vicente Martín-Sánchez, Virginia Esteve-Luque, Miguel Delgado-Rodríguez, María Ortiz-Ramos, Josep Vidal, Clotilde Vázquez, Lidia Daimiel, Emilio Ros, Cristina Razquin, Indira Paz-Graniel, Jose V. Sorlí, Olga Castañer, Antonio García-Rios, Laura Torres-Collado, Olga Fernández-Barceló, María Angeles Zulet, Elena Rayó-Gago, Rosa Casas, Naomi Cano-Ibáñez, Lucas Tojal-Sierra, Víctor J. Simón-Frapolli, Silvia Carlos, Sangeetha Shyam, Rebeca Fernández-Carrión, Albert Goday, Jose David Torres-Peña, Sandra González-Palacios, Sonia Eguaras, Nancy Babio, María Dolores Zomeño, Manuela García-de-la-Hera

**Affiliations:** 1Unidad de Epidemiología de la Nutrición (EPINUT), Departamento de Salud Pública, Historia de la Ciencia y Ginecología, Universidad Miguel Hernández (UMH), 03550 Alicante, Spain; aoncina@umh.es (A.O.-C.); lcompan@umh.es (L.C.-G.); l.torres@umh.es (L.T.-C.); sandra.gonzalezp@umh.es (S.G.-P.); manoli@umh.es (M.G.-d.-l.-H.); 2Instituto de Investigación Sanitaria y Biomédica de Alicante (ISABIAL), 03010 Alicante, Spain; 3CIBER Epidemiología y Salud Pública (CIBERESP), Instituto de Salud Carlos III (ISCIII), 28034 Madrid, Spain; abueno@ugr.es (A.B.-C.); vicente.martin@unileon.es (V.M.-S.); ocastaner@imim.es (O.C.); ncaiba@ugr.es (N.C.-I.); 4Centro de Investigación Biomédica en Red Fisiopatología de la Obesidad y la Nutrición (CIBEROBN), Institute of Health Carlos III, 28029 Madrid, Spain; mcanela@unav.es (M.R.-C.); dolores.corella@uv.es (D.C.); jordi.salas@urv.cat (J.S.-S.); mfito@imim.es (M.F.); jalfmtz@unav.es (A.M.); angelmago13@gmail.com (Á.M.A.-G.); jwarnberg@uma.es (J.W.); mariaadoracion.romaguera@ssib.es (D.R.); jlopezmir@gmail.com (J.L.-M.); restruch@clinic.cat (R.E.); fjtinahones@hotmail.com (F.J.T.); joselapetra543@gmail.com (J.L.); jalvarez@proyinves.ulpgc.es (J.Á.-P.); pep.tur@uib.es (J.A.T.); virginia.esteveluque@gmail.com (V.E.-L.); mdelgado@ujaen.es (M.D.-R.); cvazquezma@gmail.com (C.V.); lidia.daimiel@imdea.org (L.D.); eros@clinic.cat (E.R.); crazquin@unav.es (C.R.); indiradelsocorro.paz@urv.cat (I.P.-G.); jose.sorli@uv.es (J.V.S.); angarios2004@yahoo.es (A.G.-R.); olga.fb@hotmail.com (O.F.-B.); mazulet@unav.es (M.A.Z.); rcasas1@recerca.clinic.cat (R.C.); lutojal@hotmail.com (L.T.-S.); victorsimonfrapolli.med@gmail.com (V.J.S.-F.); scarlos@unav.es (S.C.); sangeetha.shyam@urv.cat (S.S.); rebeca.fernandez@uv.es (R.F.-C.); agoday@psmar.cat (A.G.); azarel_00@hotmail.com (J.D.T.-P.); seguaras@alumni.unav.es (S.E.); nancy.babio@urv.cat (N.B.); mzomeno@researchmar.net (M.D.Z.); 5IdiSNA, Department of Preventive Medicine and Public Health, University of Navarra, 31008 Pamplona, Spain; 6Department of Preventive Medicine, University of Valencia, 46010 Valencia, Spain; 7Departament de Bioquímica i Biotecnologia, Alimentació, Nutrició, Desenvolupament i Salut Mental ANUT-DSM, Universitat Rovira i Virgili, 43201 Reus, Spain; 8Institut d’Investigació Sanitària Pere Virgili (IISPV), 43007 Reus, Spain; 9Unit of Cardiovascular Risk and Nutrition, Institut Hospital del Mar de Investigaciones Médicas Municipal d’Investigació Médica (IMIM), 08003 Barcelona, Spain; 10Department of Nutrition, Food Sciences, and Physiology, Center for Nutrition Research, University of Navarra, 31008 Pamplona, Spain; 11Precision Nutrition and Cardiometabolic Health Program, IMDEA Food, CEI UAM + CSIC, 28049 Madrid, Spain; 12Bioaraba Health Research Institute, Cardiovascular, Respiratory and Metabolic Area, Osakidetza Basque Health Service, Araba University Hospital, University of the Basque Country UPV/EHU, 01009 Vitoria-Gasteiz, Spain; 13EpiPHAAN Research Group, School of Health Sciences, Instituto de Investigación Biomédica en Málaga (IBIMA), University of Málaga, 29071 Málaga, Spain; 14Health Research Institute of the Balearic Islands (IdISBa), 07120 Palma de Mallorca, Spain; elena.rayo@ssib.es; 15Department of Internal Medicine, Maimonides Biomedical Research Institute of Cordoba (IMIBIC), Reina Sofia University Hospital, University of Cordoba, 14004 Cordoba, Spain; 16Department of Internal Medicine, Institut d’Investigacions Biomèdiques August Pi Sunyer (IDIBAPS), Hospital Clinic, University of Barcelona, 08036 Barcelona, Spain; 17Institut de Recerca en Nutrició I Seguretat Alimentaria (INSA-UB), University of Barcelona, 08007 Barcelona, Spain; 18Department of Endocrinology, Instituto de Investigación Biomédica de Málaga (IBIMA), Virgen de la Victoria Hospital, University of Málaga, 29016 Málaga, Spain; 19Research Unit, Department of Family Medicine, Distrito Sanitario Atención Primaria Sevilla, 41013 Sevilla, Spain; 20Research Institute of Biomedical and Health Sciences (IUIBS), University of Las Palmas de Gran Canaria Preventive Medicine Service, Centro Hospitalario Universitario Insular Materno Infantil (CHUIMI), Canarian Health Service, 35016 Las Palmas de Gran Canaria, Spain; 21Department of Preventive Medicine and Public Health, University of Granada, 18016 Granada, Spain; 22Research Group on Community Nutrition & Oxidative Stress, University of Balearic Islands, 07122 Palma de Mallorca, Spain; 23Institute of Biomedicine (IBIOMED), University of León, 24071 León, Spain; 24Lipids and Vascular Risk Unit, Internal Medicine, Hospital Universitario de Bellvitge-IDIBELL, Hospitalet de Llobregat, 08908 Barcelona, Spain; 25Division of Preventive Medicine, Faculty of Medicine, University of Jaén, 23071 Jaén, Spain; 26Precision Nutrition and Cardiometabolic Health Program, IMDEA Alimentacion, 28049 Madrid, Spain; 27Medicine and Endocrinology, University of Valladolid, 47003 Valladolid, Spain; 28Department of Endocrinology and Nutrition, Instituto de Investigación Sanitaria Hospital Clínico San Carlos (IdISSC), 28040 Madrid, Spain; maria.ortiz.929@gmail.com; 29CIBER Diabetes y Enfermedades Metabólicas (CIBERDEM), Instituto de Salud Carlos III (ISCIII), 28029 Madrid, Spain; jovidal@clinic.cat; 30Department of Endocrinology, Institut d’Investigacions Biomèdiques August Pi Sunyer (IDIBAPS), Hospital Clinic, University of Barcelona, 08036 Barcelona, Spain; 31Department of Endocrinology and Nutrition, Hospital Fundación Jimenez Díaz, Instituto de Investigaciones Biomédicas (IISFJD), University Autonoma, 28040 Madrid, Spain; 32Nutritional Control of the Epigenome Group, Precision Nutrition and Obesity Program, IMDEA Food, CEI UAM + CSIC, 28049 Madrid, Spain; 33Departamento de Ciencias Farmacéuticas y de La Salud, Faculty de Farmacia, Universidad San Pablo-CEU, CEU Universities, 28668 Boadilla del Monte, Spain; 34Lipid Clinic, Department of Endocrinology and Nutrition, Institut d’Investigacions Biomèdiques August Pi Sunyer (IDIBAPS), Hospital Clínic, 08036 Barcelona, Spain; 35School of Health Sciences, Universitat Ramon Llull, 08025 Barcelona, Spain

**Keywords:** COVID-19, metabolic syndrome, self-reported health, Mediterranean diet, lifestyle

## Abstract

Background: The COVID-19 lockdown represented an immense impact on human health, which was characterized by lifestyle and dietary changes, social distancing and isolation at home. Some evidence suggests that these consequences mainly affected women and altered relevant ongoing clinical trials. The aim of this study was to evaluate the status and changes in diet, physical activity (PA), sleep and self-reported health status (SRH) as perceived by older adult men and women with metabolic syndrome during the COVID-19 lockdown. Methods: We analyzed data from 4681 Spanish adults with metabolic syndrome. We carried out a telephone survey during May and June 2020 to collect information on demographics, dietary habits, PA, sleep, SRH and anthropometric data. Results: The mean age of participants was 64.9 years at recruitment, and 52% of participants were men. Most participants (64.1%) perceived a decrease in their PA during confinement. Regarding gender-specific differences, a higher proportion of women than men perceived a decrease in their PA (67.5% vs. 61.1%), Mediterranean diet adherence (20.9% vs. 16.8%), sleep hours (30.3% vs. 19.1%), sleep quality (31.6% vs. 18.2%) and SRH (25.9% vs. 11.9%) (all *p* < 0.001). Conclusions: The COVID-19 lockdown affected women more negatively, particularly their self-reported diet, PA, sleep and health status.

## 1. Introduction

The COVID-19 pandemic was the greatest challenge faced by Public Health agencies in recent times [1]. Due to the high contagiousness of the virus and the associated high mortality rate, several virus containment measures were decreed worldwide, the most effective being the population lockdown [2]. This new situation resulted in the population around the world remaining in isolation for a considerable period of time, sometimes alone, leading to different health consequences, particularly adverse psychological effects such as anxiety [3,4]. It also led to unhealthy lifestyle habits such as eating diets of low nutritional quality [5], oversleeping or undersleeping [6,7].

However, these health consequences were perceived differently depending on the country studied. For instance, a previous study showed that Spain was one of the countries with the highest risk perception and psychological distress related to COVID-19, second only to the UK [8]. In Spain, the lockdown began on 15 March 2020, along with other measures such as wearing masks and social distancing [9]. The isolation and loneliness resulting from these measures particularly affected the elderly and/or those at high risk of infection, such as people with chronic diseases, and in some cases, even led to a higher mortality rate in these groups [4,10,11,12]. A recent study in a Spanish population with type 2 diabetes showed that during the lockdown, although unhealthy lifestyle habits, such as decreased physical activity (PA) and increased consumption of sugary food and snacks, were acquired, the consumption of vegetables increased [13]. Similar results were obtained in Spanish adults at high risk of developing type 2 diabetes, which showed an increase in adherence to the Mediterranean diet (MedDiet) and a reduction in the body mass index (BMI) during lockdown, despite a decrease in their level of PA, sleep quality and general health [14].

Differences in lifestyles between men and women could result in different health effects of lockdown. However, there are few studies that have evaluated lifestyle among Spanish men and women. One study showed no differences between men and women in terms of self-reported health (SRH) quality [14]. A cross-sectional study that included 72 adults with type 2 diabetes has shown that women reported more food cravings than men [13], which can be associated with low-quality diets [15], increased weight gain [16] and emotional alterations [17]. Another cross-sectional study examined data from 3041 individuals aged ≥65 years with chronic diseases during the lockdown and reported that men showed a higher risk of adopting unhealthier lifestyles, such as lower PA levels and higher rates of sedentariness [18]. Contrary, and according to the findings of an online survey conducted with 3480 participants, women exhibited a higher vulnerability to mental disorders, specifically anxiety, during the lockdown period [19].

Previous studies suggest that lockdowns may have different effects on the lifestyles and health of the population, with particularly negative effects on older adults with chronic diseases [20,21]. This population has a heightened vulnerability to the effects of confinement compared to other age groups [22], mainly because they face an increased risk of severe complications from infectious diseases like COVID-19, attributed to their compromised immune systems and prevalent pre-existing medical conditions [23]. Moreover, prolonged social isolation, restrictions on PA and limited access to medical services during lockdown can significantly impact their mental health, increasing the risk of depression, anxiety and cognitive decline [24], as well as diminishing their quality of life [25,26]. There is also evidence of different effects of lockdown in men and women, although results are still inconclusive and contradictory [27]. Thus, the aim of this study was to evaluate the status and the changes in the adherence to MedDiet, PA, sleep and SRH as perceived by men and women with metabolic syndrome (MetS) participating in an intervention study during the COVID-19 lockdown. In addition, since there is also some evidence that the COVID-19 lockdown could affect the correct development of ongoing clinical trials [28], we will also assess if the lockdown affected participants in the control and the intervention group differently.

## 2. Materials and Methods

### 2.1. Study Design and Population

A cross-sectional analysis was conducted within the framework of the PREDIMED-Plus study (Spain) (www.predimedplus.es, accessed on 29 May 2024). PREDIMED-Plus is an ongoing multicenter (23 Spanish centers) parallel-group, 8-year intervention study, which consisted of 6 years of active intervention and 2 years of follow-up without intervention. The main objective of this intervention study is to evaluate the effect of an intensive intervention [weight loss via the consumption of an energy-reduced MedDiet, promotion of PA and behavioral support] compared to a control group [usual care with non-energy-reduced MedDiet advice] on the primary prevention of CVD and mortality in adults with high cardiometabolic risk.

The recruitment of participants took place between October 2013 and December 2016. Participants were randomized in a 1:1 ratio to one of the two intervention groups. The study populations were men (55–75 years) and women (60–75 years) with overweight or obesity (BMI 27–40 kg/m^2^), free of CVD, who met at least three criteria of MetS according to the updated criteria of the International Diabetes Federation and the American Heart Association and National Heart, Lung and Blood Institute [29]. PREDIMED-Plus used a BMI threshold of 27 kg/m^2^ for categorizing overweight, which differs from the WHO’s standard cut-off of 25 kg/m^2^ [30], to better capture cardiovascular risk within their specific population [31,32]. More detailed information regarding the study design and methods has been published previously [33].

The study protocol was registered at the International Standard Randomized Controlled Trial (ISRCTN: http://www.isrctn.com/ISRCTN89898870, accessed on 3 June 2024), implemented following the ethical standards of the Declaration of Helsinki and approved by the Research Ethics Committees from all of the 23 recruiting centers. All participants provided written informed consent.

A total of 6874 participants were randomized to the PREDIMED-Plus study. Out of the 6686 participants with complete information at the baseline visit, 5682 responded to the COVID-19 telephone survey. Among these, 1001 participants had incomplete information for the questions of interest in this study regarding MedDiet, PA, sleep and SRH. Thus, the final sample of our study was 4681 participants (Figure 1).

At the time of the COVID-19 lockdown in Spain, the PREDIMED-Plus study was in the middle of the intervention (fourth and fifth year from the beginning). Thus, the recruitment of participants was completed, and participants in the intervention group were receiving advice on weight loss or maintaining weight loss when the pandemic broke out.

### 2.2. Data Collection

From 15 March to 20 June 2020, like the rest of the general population in Spain, the study participants were exposed to a period of restricted household confinement. In order to know the characteristics and conditions under which the participants had experienced the lockdown, we designed an ad-hoc questionnaire (Appendix A). Questions included in this questionnaire were formulated by the PREDIMED-Plus study node coordinators after receiving feedback from all principal investigators. It should be noted that these questions were specifically designed to address the objective of the present study and were collected in parallel with the information gathered per protocol in the PREDIMED-Plus study. This questionnaire was managed through the PREDIMED-Plus intranet, which is the communication channel used by all the centers participating in the study to digitize the information. During May and June 2020, trained fieldworkers administered a 24-item ad-hoc telephonic questionnaire, which included questions on the following: sociodemographic data, characteristics and conditions of the personal confinement situation, SRH and lifestyles (MedDiet, PA, sleep). Information on the degree of compliance with household confinement was not collected because it was a mandatory measure of containment for the Spanish population that was closely monitored by the state security forces.

### 2.3. Lifestyles, SRH and Other Variables

We used the questionnaire mentioned above to assess SRH and different lifestyle aspects, such as adherence to the MedDiet, PA and sleep. To collect information on these variables, we used closed-ended questions referring to two timeframes: during confinement and a comparison of the situation before and after confinement. The second timeframe questions were aimed to assess any changes that the participants underwent during the confinement. Most of the questions could be answered on a 5-point Likert scale with answer choices ranging from “excellent” to “poor”. This kind of scale, although not strictly validated, is widely used in the scientific literature focused on analyzing sleep quality [34,35] and/or SRH [36,37]. In the present study, we also described other variables such as intervention group, age, sex, as well as reported weight and height, which were used to calculate BMI. 

Questions included in the self-developed questionnaire that we used can be consulted in Appendix A.

### 2.4. Statistical Analysis

For the present study, we used the PREDIMED-Plus database updated until July 2021. We used one-factor ANOVA for quantitative variables and chi-square tests for categorical variables to assess the differences in the baseline characteristics between respondents and non-respondents to the COVID-19 survey in the PREDIMED-Plus sample. 

We described each response to questions using the number (*n*) and percentage (%). To compare the differences in response proportions between men and women as well as between intervention groups, we performed bivariate analysis (chi-square test). Since very few participants reported the highest or lowest categories, such as ‘very good’ or ‘very bad’, we combined the categories, creating three-category variables in some questions (Appendix A).

Statistical analysis was performed using the software R version 4.0.3, with a significance threshold of α = 0.05 applied to define associations as statistically significant.

## 3. Results

### 3.1. General Sample Characteristics

Table 1 shows the baseline characteristics of the total sample of PREDIMED-Plus, distinguishing between participants who responded to the COVID-19 survey and those who did not. A total of 4681 participants completed the survey, 51% of whom were from the control group. In comparison to the participants who did not respond to the COVID-19 survey, those who did respond showed higher levels of PA, increased alcohol consumption, a lower prevalence of both self-reported hypercholesterolemia and type 2 diabetes, as well as a higher proportion of good SRH.

### 3.2. Status and Changes in MedDiet, PA, Sleep and SRH Perceived by All Participants during the COVID-19 Lockdown

In general, 72.6% of participants reported that their compliance with the MedDiet recommended in the study did not change during confinement (Table 2). Most participants (64.1%) indicated that their level of PA decreased during confinement. Almost half of the participants (43.5%) reported sleeping 6–7 h during confinement. In addition, the majority reported having an average sleep quality (50.8%) during confinement and reported that their sleep quality had not changed (74.3%). Overall, participants had a good SRH (63.1%) during confinement, similar to their usual status (79.5%).

### 3.3. Status and Changes in MedDiet, PA, Sleep and SRH Perceived by Men and Women during the COVID-19 Lockdown

Women were reported to be more negatively affected by the lockdown than men (Table 2). A higher number of women reported worsening of MedDiet compliance (a little/much worse: 20.9% women vs. 16.8% men, *p* < 0.001). Women also reported a lower level of PA (67.5% in women vs. 61.1% in men, *p* < 0.001). A higher percentage of women reported sleeping 6 h or less per day (30.3% vs. 19.1% men; *p* < 0.001). More than 30% of women reported a fairly/very bad (31.6% vs. 18.2% men *p* < 0.001) sleep quality during lockdown as well as a worse than usual sleep quality (25.9% vs. 16.9% men; *p* < 0.001). In addition, 25.9% (11.9% men) (*p* < 0.001) of women indicated fair/poor SRH during lockdown. Finally, a higher percentage of women reported a little/much worse SRH (18.4% women vs. 10.2% men, *p* < 0.001) during lockdown than before confinement.

### 3.4. Status and Changes in MedDiet, PA, Sleep and SRH Perceived by Participants in Control and Intervention Groups during the COVID-19 Lockdown

We observed statistically significant differences in PA and SRH. A higher proportion of participants in the control group (64.6%) reported a decrease in their level of PA during lockdown than in the intervention group (63.6%) (*p* = 0.011) (Table 3). The majority of participants in both groups reported good SRH during lockdown, although the percentage was higher in the control group (65.1% vs. 61.0% intervention group; *p* = 0.028). In contrast, a higher percentage of participants in the intervention group reported very good/excellent SRH (19.5% vs. 17.1% control group; *p* = 0.028). A higher proportion of participants in the intervention group indicated that their SRH improved (7.3% vs. 5.6% control group) or worsened (14.6% vs. 13.6% control group) during lockdown compared to their usual SRH (*p* = 0.022).

## 4. Discussion

In this study, we have shown that overall, participants reported no changes during lockdown in their compliance with MedDiet study recommendations, sleep quality, hours of sleep per day or SRH, although they reported a decrease in their PA levels. When we stratified by gender, a negative self-perceived effect of confinement on MedDiet, PA, sleep and SRH was mainly observed in women. In addition, when we stratified by study group, we observed a greater decrease in PA levels in the control group participants.

Self-perceived PA level was the most affected by confinement among the PREDIMED-Plus study participants. This is not an uncommon finding since the decrease in the level of PA during confinement has been evidenced in different populations, such as university students [38] or older adults [39]. An international survey involving thirty-five research organizations from Europe, North Africa, Western Asia and the Americas was conducted in April 2020 to assess changes in adult PA and dietary habits before and during the COVID-19 lockdown [40]. This electronic survey was answered by 1047 adults, mostly women (54%), and the results showed a negative effect on vigorous, moderate, walking and overall PA levels. Similar results were found in a survey carried out in Spain [31] with 3800 healthy adults who were asked about their perceived level of PA. The results showed that self-reported PA decreased significantly during confinement in their study population. Moreover, in a Spanish cohort of adults with type 2 diabetes, a study population that was very similar to ours, also found a high percentage of physical inactivity during the COVID-19 lockdown [13] that was attributed to the severe lockdown measures. Social distancing, limitations on group gatherings and PA restrictions in open spaces abruptly altered the traditional lifestyle, leading to potential consequences on the psychological and emotional states of the population.

We would like to point out that our results regarding PA levels during the COVID-19 lockdown differ slightly from those found in a study with the same participants and over a similar time period [41]. In this study, Paz-Graniel et al. assessed the effect of lockdown restrictions on components of PREDIMED-Plus intervention, such as diet and PA, with data obtained during three lockdown phases (pre-lockdown, proper lockdown and post-lockdown). In line with our results, they found a non-significant decrease in PA during lockdown compared to pre-lockdown (−2.60 min metabolic equivalent task minutes (METs·min/day), *p* > 0.05) as well as a significant decrease in intense PA during lockdown (−15.4 METs·min/day, *p* < 0.01) [41]. However, in contrast to our findings, they found a significant increase in moderate PA during lockdown (12.1 METs·min/day, *p* < 0.01) [41]. The differences between the results of the two studies can be attributed to several key factors. Firstly, there is a notable disparity in the methods used to assess PA. Paz-Graniel et al. [41] used validated questionnaires that not only determined overall PA but also classified physical activities into distinct categories such as light, moderate and intense. In contrast, our study relied on a single self-reported question to evaluate PA, which may have resulted in less accurate results regarding participants’ activity levels. Secondly, although both studies focused on the COVID-19 lockdown, the timeframes studied were different. While the first study set the proper lockdown period as March to December 2020, we focused our survey on a much more specific timeframe (May–June 2020), which was a period marked by the imposition of strict lockdown measures. This discrepancy in timeframes may account for differing patterns in PA levels. Thus, each study reflected different stages of evolving circumstances and adherence to lockdown measures during the specified months, which could have affected PA levels.

In our study, more women than men reported a worsening of sleep and SRH, as well as less PA and lower adherence to MedDiet during lockdown. In line with these results, cross-sectional studies carried out within the framework of the COVID-19 lockdown have shown greater negative lifestyle consequences derived from confinement in women. A survey carried out on a Danish population showed that women increased their consumption of pastries and also their total food consumption during confinement more than men [42]. A multicentre and international cross-sectional study performed on 22,330 adults from thirteen different countries has shown that, during the first wave of the COVID-19 pandemic, women were more likely to suffer from insomnia disorders than men [43]. Moreover, a study carried out in Arkansas showed that men spent 30 min more PA per day than women during confinement [44]. In addition to PA, the authors assessed SRH, but contrary to our findings, they found no statistically significant differences between men and women [44]. These contradictory results may be due to various factors, such as participants’ age, health conditions and country studied, although there is no clear explanation. However, general differences in lifestyle factors between men and women could be partially explained by women’s increased demands at home during confinement compared to men [45]. In other words, women’s increased housework responsibilities during the COVID-19 pandemic may have reduced their leisure time and/or opportunity to practice PA.

There is strong evidence highlighting the gender inequalities experienced during the COVID-19 lockdown. In Germany, one study carried out during the confinement showed that women were more seriously affected than men, not only in physical terms but also in cognitive ones [45]. In the United Kingdom, a survey of an adult population during the COVID-19 lockdown showed that women spent more time on unpaid work like housework and childcare rather than leisure activities, which resulted in greater psychological distress [46]. We should point out that there were very few studies in which the results were differentiated between men and women. We have found a possible justification for this in the bibliometric study carried out by Jimenez Carrillo et al. with respect to articles about COVID-19 in Spain [47]. This study showed that fewer women were listed as first authors of these articles than men, showing gender inequalities in authorship. In addition, only 1% of those mentioned in the study stratified the results by gender, and the majority of these listed a woman as the first author.

We observed some differences between participants in the control group and those in the intervention group in our study. On the one hand, we observed that participants in the control group reduced their levels of PA during confinement to a greater extent than those in the intervention group. This may be because participants in the intervention group had previously received three years of counseling on MedDiet and PA, which may have increased their awareness of the importance and benefits of PA. Another explanation could be that participants in the control group had a higher BMI than those in the intervention group [48], a factor that has been associated with lower levels of PA and poorer diet quality in lockdown [49]. On the other hand, participants in the control group reported good SRH during confinement to a greater extent than those in the intervention group. However, a higher percentage of participants in the intervention group reported a very good SRH. A higher level of PA during confinement has been associated with better well-being [50,51]. This could partially justify the fact that a higher proportion of participants in the intervention group reported a very good SRH if we consider that they reported a higher level of PA in lockdown.

There are several limitations in our study that we should mention. Our study sample included Spanish adults with MetS, which could reduce the generalizability of our results, especially in younger populations. Additionally, after examining the characteristics of our sample, we observed that participants who responded to the COVID-19 survey had a healthier lifestyle, with a higher degree of PA and a lower incidence of chronic diseases such as high cholesterol and type 2 diabetes than those PREDIMED-plus participants who did not respond to this specific questionnaire. This health awareness could explain their greater participation in the survey. The tool we used to collect the information for this study is an unvalidated self-elaborated survey. Thus, our results could have questionable validity and reliability. However, we want to point out that the survey we used was developed by an expert committee and included questions widely used in the scientific literature. This survey was self-reported and, therefore, the answers depended on the participant’s understanding of the questions, which may have led to response biases and especially desirability bias. We had to carry out the survey within a specific timeframe, and in order to collect data on multiple topics as quickly as possible, we had to limit some variables to a single question and a fixed time period. For example, the survey did not include questions about specific foods, PA type, or social interactions. Nor did we include information on negative lifestyle factors such as sitting time in the survey. We collected information in relation to two timeframes (before and during confinement), but our descriptive cross-sectional design did not allow us to make temporal or causal assumptions about the observed results.

Some strengths also characterized our study. The study sample was large and representative of Spanish adults with MetS. This is a novel study because, as far as we know, there are no other published studies that have assessed the self-perceived changes in different lifestyle factors among participants in a dietary intervention study during lockdown. The survey that we used was carried out in a short critical period of the epidemic in Spain, not many weeks after the lockdown began. That gave us the opportunity to examine the situation with a more realistic approach and minimize the recall bias. In some way, our results indicate that the PREDIMED-Plus intervention study is achieving its purpose. In this sense, we have observed that most of the participants reported not having changed their MedDiet adherence during confinement, which has been rarely reported in previous articles [52]. This fact could well be attributed to the support provided by the PREDIMED-Plus fieldworkers to the participants and the empowerment of participants due to the knowledge acquired during the intervention [53]. Lastly, our results provide new evidence regarding gender inequalities during the COVID-19 pandemic, an important and understudied topic.

## 5. Conclusions

We have provided additional evidence on the lifestyle and self-reported health of an adult Spanish population during the COVID-19 lockdown. Overall, participants only perceived a worsening in their PA levels during lockdown. However, when we analyzed the results by gender, we observed statistically significant differences between men and women. The lockdown affected women negatively to a greater extent. However, as the consequences of the COVID-19 pandemic are still ongoing, it would be very interesting to compare our results with those observed in the framework of other intervention studies that were being carried out before, during and after the COVID-19 lockdown.

## Figures and Tables

**Figure 1 nutrients-16-02000-f001:**
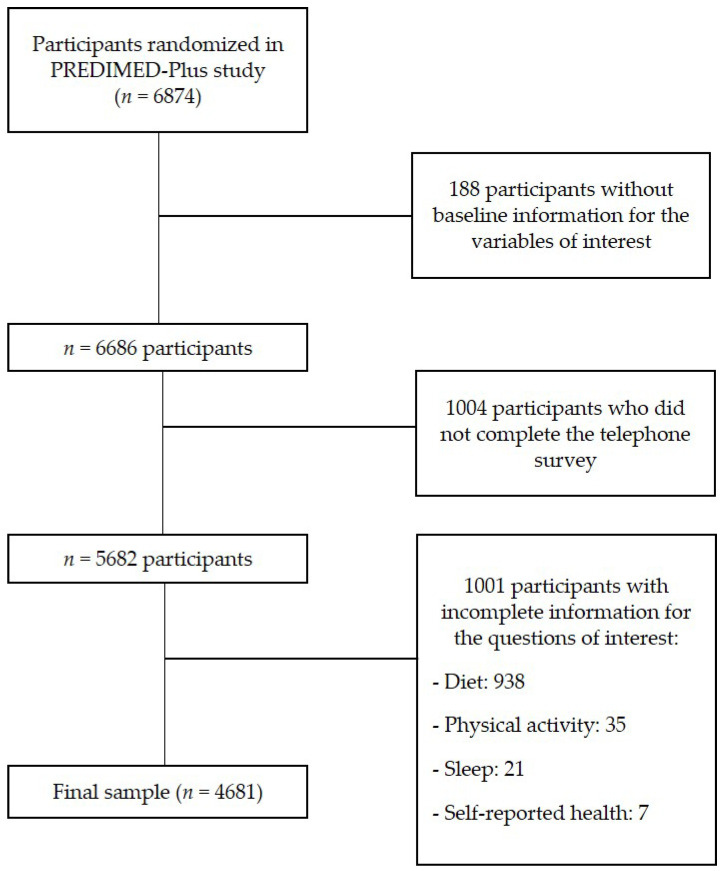
Flowchart of participants included in the present analysis from the PREDIMED-Plus Study.

**Table 1 nutrients-16-02000-t001:** Baseline characteristics and lifestyle among participants in the PREDIMED-Plus study and COVID-19 sub-study.

Variables	Total Sample (*n* = 6686)	COVID-19 Survey		*p*-Value ^1^
		Participants (*n* = 4681)	Non-Participants (*n* = 2005)	
Age at recruitment ^2^, years (SD)	64.9 (4.9)	65.0 (4.9)	64.9 (5.0)	0.792
Intervention group, *n* (%)				0.321
Intervention	3306 (49.4)	2296 (49.0)	1010 (50.4)	
Control	3380 (50.6)	2385 (51.0)	995 (49.6)	
Sex, *n* (%)				0.196
Male	3442 (51.5)	2434 (52.0)	1008 (50.3)	
Female	3244 (48.5)	2247 (48.0)	997 (49.7)	
Educational level, *n* (%)				0.388
Illiterate or primary	3262 (48.8)	2291 (48.9)	971 (48.4)	
Secondary	1939 (29.0)	1371 (29.3)	568 (28.3)	
Academic or graduate	1485 (22.2)	1019 (21.8)	466 (23.2)	
BMI ^2^, kg/m^2^ (SD)	32.6 (3.4)	32.5 (3.5)	32.7 (3.4)	0.084
Alcohol consumption ^2^, g/day (SD)	11.2 (15.3)	11.5 (15.6)	10.4 (14.6)	0.005
Smoking status, *n* (%)				0.152
Current	864 (12.9)	583 (12.5)	281 (14.0)	
Former	2901 (43.4)	2027 (43.3)	874 (43.6)	
Never	2921 (43.7)	2071 (44.2)	850 (42.4)	
Adherence to Mediterranean diet ^2,3^, 0–17 points (SD)	8.5 (2.7)	8.5 (2.7)	8.4 (2.7)	0.070
Physical activity ^2,4^, METS-min/day (SD)	353.6 (329.8)	367.6 (335.7)	320.9 (313.5)	<0.001
Sleep (hours/day), *n* (%)				0.285
<6	2205 (33.0)	1542 (32.9)	663 (33.1)	
6–7	2143 (32.1)	1481 (31.6)	662 (33.0)	
8–9	2154 (32.2)	1519 (32.5)	635 (31.7)	
>9	184 (2.8)	139 (3.0)	45 (2.2)	
Self-reported health status, *n* (%)				0.009
Excellent	74 (1.1)	54 (1.2)	20 (1.0)	
Very good	579 (8.7)	398 (8.5)	181 (9.0)	
Good	3754 (56.2)	2694 (57.6)	1062 (53.0)	
Fair	2094 (31.3)	1411 (30.1)	683 (34.1)	
Poor	183 (2.7)	124 (2.6)	59 (2.9)	
Disease prevalence ^5^, *n* (%)				
Hypertension	5563 (83.2)	3897 (83.3)	1666 (83.1)	0.959
High blood cholesterol	4646 (69.5)	3171 (67.7)	1475 (73.6)	<0.001
Type 2 Diabetes	2039 (30.5)	1359 (29.0)	680 (33.9)	<0.001

Abbreviations: BMI, body mass index; SD, standard deviation. ^1^ *p*-value (*p*) from chi-squared test (categorical variables) and ANOVA (continuous variables). ^2^ Mean (SD). ^3^ Adherence to an energy-restricted MedDiet was assessed using a 17-item questionnaire, a modified version of a validated 14-item questionnaire. ^4^ MET-min, metabolic-equivalent task minutes. ^5^ Self-reported answers.

**Table 2 nutrients-16-02000-t002:** Results of the survey on COVID-19 according to gender in 4681 participants of the PREDIMED-Plus study.

			Gender, *n* (%)		*p*-Value ^1^
		Total (*n* = 4681)	Men 2434 (51.9)	Women 2247 (48.1)	
MedDiet					
Do you think your adherence to the healthy Mediterranean diet that we are recommending in this study has changed during your confinement?	Improved a lot/a little	405 (8.6)	249 (10.2)	156 (7.0)	
	Not changed	3399 (72.6)	1777 (73.0)	1622 (72.2)	
	A little/much worse	877 (18.7)	408 (16.8)	469 (20.9)	<0.001
PA					
Has your physical activity level changed during confinement?	More physical activity	428 (9.1)	269 (11.1)	159 (7.1)	
	Less physical activity	3002 (64.1)	1487 (61.1)	1515 (67.5)	
	No	1251 (26.7)	678 (27.9)	573 (25.5)	<0.001
Sleep					
On average, how many hours did you sleep at night during confinement?	Less than 6 h	1146 (24.5)	466 (19.1)	680 (30.3)	
	6–7 h	2035 (43.5)	1133 (46.5)	902 (40.1)	
	8–9 h	1295 (27.7)	729 (30.0)	566 (25.2)	
	More than 9 h	205 (4.4)	106 (4.4)	99 (4.4)	<0.001
How would you rate the quality of your sleep during confinement?	Very/Fairly good	1153 (24.6)	654 (26.9)	499 (22.2)	
	Average	2377 (50.8)	1337 (54.9)	1040 (46.3)	
	Fairly/Very bad	1151 (24.5)	443 (18.2)	708 (31.6)	<0.001
How would you rate your sleep quality during confinement compared to your usual sleep quality?	Better than usual	213 (4.6)	93 (3.8)	120 (5.3)	
	Worse than usual	992 (21.2)	411 (16.9)	581 (25.9)	
	Same as usual	3476 (74.3)	1930 (79.3)	1546 (68.8)	<0.001
SRH					
How would you rate your health during confinement?	Excellent/Very good	855 (18.3)	514 (21.1)	341 (15.2)	
	Good	2953 (63.1)	1630 (67.0)	1323 (58.9)	
	Fair/Poor	873 (18.7)	290 (11.9)	583 (25.9)	<0.001
How would you say your health is now compared to before confinement?	Much/A little better	301 (6.4)	158 (6.5)	143 (6.3)	
	About the same	3720 (79.5)	2028 (83.3)	1692 (75.3)	
	A little/Much worse	660 (14.1)	248 (10.2)	412 (18.4)	<0.001

Abbreviations: PA, physical activity; SRH, self-reported health. ^1^ *p*-value corresponding to the chi-square test to compare responses according to gender.

**Table 3 nutrients-16-02000-t003:** Results of the survey on COVID-19 according to intervention group in 4681 participants of the PREDIMED-Plus study.

			Group, *n* (%)		*p*-Value ^1^
		Total (*n* = 4681)	Control 2385 (51.0)	Intervention 2296 (49.0)	
MedDiet					
Do you think your adherence to the healthy Mediterranean diet that we are recommending in this study has changed during your confinement?	Improved a lot/a little	405 (8.6)	201 (8.4)	204 (8.9)	
	It has not changed	3399 (72.6)	1766 (74.0)	1633 (71.1)	
	A little/much worse	877 (18.7)	418 (17.5)	459 (20.0)	0.114
PA					
Has your physical activity level changed during confinement?	More physical activity	428 (9.1)	189 (7.9)	239 (10.4)	
	Less physical activity	3002 (64.1)	1541 (64.6)	1461 (63.6)	
	No	1251 (26.7)	655 (27.5)	596 (26.0)	0.011
Sleep					
On average, how many hours did you sleep at night during confinement?	Less than 6 h	1146 (24.5)	600 (25.2)	546 (23.8)	
	6–7 h	2035 (43.5)	1031 (43.2)	1004 (43.7)	
	8–9 h	1295 (27.7)	636 (26.7)	659 (28.7)	
	More than 9 h	205 (4.4)	118 (4.9)	87 (3.8)	0.097
How would you rate the quality of your sleep during confinement?	Very/Fairly good	1153 (24.6)	594 (24.9)	559 (24.4)	
	Average	2377 (50.8)	1223 (51.3)	1154 (50.3)	
	Fairly/Very bad	1151 (24.5)	568 (23.8)	583 (25.4)	0.431
How would you rate your sleep quality during confinement compared to your usual sleep quality?	Better than usual	213 (4.6)	111 (4.7)	102 (4.4)	
	Worse than usual	992 (21.2)	482 (20.2)	510 (22.2)	
	Same as usual	3476 (74.3)	1792 (75.1)	1684 (73.3)	0.242
SRH					
How would you rate your health during confinement?	Excellent/Very good	855 (18.3)	407 (17.1)	448 (19.5)	
	Good	2953 (63.1)	1553 (65.1)	1400 (61.0)	
	Fair/Poor	903 (18.8)	425 (17.9)	448 (19.5)	0.028
How would you say your health is now compared to before confinement?	Much/A little better	301 (6.4)	134 (5.6)	167 (7.3)	
	About the same	3720 (79.5)	1928 (80.8)	1792 (78.0)	
	A little/Much worse	660 (14.1)	323 (13.6)	337 (14.6)	0.022

Abbreviations: PA, physical activity; SRH, self-reported health. ^1^ *p*-value corresponding to the chi-square test to compare responses according to intervention group.

## Data Availability

There are restrictions on data availability for the PREDIMED-Plus trial due to the signed consent agreements around data sharing, which only allow access to external researchers for studies following the project’s purposes. Requestors wishing to access the PREDIMED-Plus trial data used in this study can make a request to the PREDIMED-Plus trial Steering Committee chair: jordi.salas@urv.cat. The request will then be passed to members of the PREDIMED-Plus Steering Committee for deliberation.

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
