# Peer review of "More Adult Women than Men at High Cardiometabolic Risk Reported Worse Lifestyles and Self-Reported Health Status in the COVID-19 Lockdown"

_nutrients, 2024, doi:10.3390/nu16132000_

Round 1
Reviewer 1 Report
Comments and Suggestions for Authors
Line 137- 140 page 3
Find a citation about the potential side effects of the SARS-CoV-2 Pandemic
From Line 148 page 3
Materials and MethodsàStudy Design and Population
The method described is relevant to the main PREDIMED-Plus study but lacks a detailed explanation of the methodology used to collect data for the stated objective: "Thus, the aim of this study was to evaluate the status and the changes in the adherence to MedDiet, PA, sleep, and SRH as perceived by men and women with metabolic syndrome (MetS) participating in an intervention study during the COVID-19 lockdown." Please provide more specific details on the data collection methods
Author Response
We appreciate the reviewers’ comments and the editor for giving us the opportunity to improve the manuscript. Below, we respond to the specific points raised by the reviewers.
Line 137- 140 page 3: Find a citation about the potential side effects of the SARS-CoV-2 Pandemic
Thank you very much for your comment. In accordance with it, we have added some new relevant citations for each sentence at the start of the last paragraph in the Introduction.
New references (page 3, lines 139 and 147):
- Rodríguez-González, R.; Facal, D.; Martínez-Santos, A.-E.; Gandoy-Crego, M. Psychological, Social and Health-Related Challenges in Spanish Older Adults During the Lockdown of the COVID-19 First Wave. Front. Psychiatry 2020, 11, 588949, doi:10.3389/fpsyt.2020.588949.
- Rodríguez-Gómez, I.; Sánchez-Martín, C.; García-García, F.J.; García-Esquinas, E.; Miret, M.; Vicente-Rodriguez, G.; Gusi, N.; Mañas, A.; Carnicero, J.A.; Gonzalez-Gross, M.; et al. The Medium-Term Consequences of a COVID-19 Lockdown on Lifestyle among Spanish Older People with Hypertension, Pulmonary Disease, Cardiovascular Disease, Musculoskeletal Disease, Depression, and Cancer. Epidemiol Health 2022, 44, e2022026, doi:10.4178/epih.e2022026.
- Fenollar-Cortés, J.; Jiménez, Ó.; Ruiz-García, A.; Resurrección, D.M. Gender Differences in Psychological Impact of the Confinement During the COVID-19 Outbreak in Spain: A Longitudinal Study. Front. Psychol. 2021, 12, 682860, doi:10.3389/fpsyg.2021.682860.
From Line 148 page 3: Materials and Methods’ Study Design and Population
The method described is relevant to the main PREDIMED-Plus study but lacks a detailed explanation of the methodology used to collect data for the stated objective: "Thus, the aim of this study was to evaluate the status and the changes in the adherence to MedDiet, PA, sleep, and SRH as perceived by men and women with metabolic syndrome (MetS) participating in an intervention study during the COVID-19 lockdown." Please provide more specific details on the data collection methods
Thank you for this appreciation. The information in the "Data Collection" section refers only to the questionnaires and information collected for the present work, which is part of the PREDIMED-Plus study. However, we agree that this point was not sufficiently clear in the text, so we have specified it better. Additionally, we have indicated in this section the availability of Table S1 so that readers can consult the questionnaire items if needed.
New text (page 5, lines 216-218):
It should be noted that these questions were specifically designed to address the objective of the present study and were collected in parallel with the information gathered per protocol in the PREDIMED-Plus study.
Thank you very much for your comments. We believe they have been very helpful in improving the understanding of the article.

Reviewer 2 Report
Comments and Suggestions for Authors
A very interesting publication :) I would only have a few suggestions.
You have qualified men 55-75 years old, women 60-75 years old.... It is worth mentioning in the Introduction why a group with this age is important in this study and not, for example, young people.
Many definitions include the word "sex". It should be "gender".
line 160: why is overweight from 27 kg/m2 and not from 25 kg/m2? Why didn't you add obesity above 40 kg/m2?
Table 1. What are the results in brackets? Percentages? It is worth adding it, e.g. (%) next to the words "total sample", "participants" and "non-participants".
Author Response
We appreciate the reviewers’ comments and the editor for giving us the opportunity to improve the manuscript. Below, we respond to the specific points raised by the reviewers.
A very interesting publication :) I would only have a few suggestions.
We really appreciate this comment.
You have qualified men 55-75 years old, women 60-75 years old.... It is worth mentioning in the Introduction why a group with this age is important in this study and not, for example, young people.
Thank you. We have included new information and references.
New text (page 3, lines 137-146):
Previous studies suggest that lockdowns may have different effects on the lifestyles and health of the population, with particularly negative effects on older adults with chronic diseases [20,21]. This population has a heightened vulnerability to the effects of confinement compared to other age groups [22], mainly because they face an increased risk of severe complications from infectious diseases like COVID-19, attributed to their com-promised immune systems and prevalent pre-existing medical conditions [23]. Moreover, prolonged social isolation, restrictions on physical activity, and limited access to medical services during lockdown can significantly impact their mental health, increasing the risk of depression, anxiety, and cognitive decline [24], as well as diminishing their quality of life [25,26].
Many definitions include the word "sex". It should be "gender".
Thank you for this appreciation. Following your comment, we have replaced the word “sex” for the word “gender” throughout the text.
line 160: why is overweight from 27 kg/m2 and not from 25 kg/m2? Why didn't you add obesity above 40 kg/m2?
Thank you for this comment. The main objective of PREDIMED-Plus is the prevention of cardiovascular diseases in a population at high cardiovascular risk. Previous scientific evidence supports that cardiovascular risk particularly increases with a BMI of 27 kg/m2 (Iyen B et al, 2021 DOI: 10.1186/s12889-021-10606-1; Adams B et al, 2020 DOI: 10.1186/s12872-020-01542-w). Thus, PREDIMED-Plus used a BMI threshold of 27 kg/m2 for categorizing overweight, which differs from the WHO's standard cut-off of 25 kg/m2, to better capture cardiovascular risk within their specific population.
Similarly, to better address and study the specific cardiovascular risks pertinent to their cohort, participants with morbid obesity (BMI above 40 kg/m2) were not included. Morbid obesity carries a higher risk of severe medical complications such as cardiovascular diseases, type 2 diabetes, hypertension, among others. Dietary and exercise interventions, such as those carried out in PREDIMED-Plus, may not be suitable or safe without extensive medical supervision in these cases.
We understood these questions as a personal clarification from the reviewer, so we have not added this justification to the manuscript and referred to the reference from PREDIMED-PLUS study profile. In case the reviewer finds it necessary, we will be happy to include this information in the article.
Table 1. What are the results in brackets? Percentages? It is worth adding it, e.g. (%) next to the words "total sample", "participants" and "non-participants".
Thank you for this comment. We have better clarified what information is included in brackets by indicating it next to each variable (Table 1, page 7).
Thank you for your pertinent questions and revision, we are sure they have helped us to improve the article.
